# Implementation of a Six-Layer Smart Factory Architecture with Special Focus on Transdisciplinary Engineering Education

**DOI:** 10.3390/s21092944

**Published:** 2021-04-22

**Authors:** Benjamin James Ralph, Marcel Sorger, Benjamin Schödinger, Hans-Jörg Schmölzer, Karin Hartl, Martin Stockinger

**Affiliations:** Chair of Metal Forming, Montanuniversität Leoben, Franz Josef Str. 18, 8700 Leoben, Austria; marcel.sorger@unileoben.ac.at (M.S.); benjamin.schoedinger@unileoben.ac.at (B.S.); hj.schmo@gmail.com (H.-J.S.); karin.hartl@unileoben.ac.at (K.H.); martin.stockinger@unileoben.ac.at (M.S.)

**Keywords:** engineering education, smart factory, digitalization, industry 4.0, metal processing, layer architecture

## Abstract

Smart factories are an integral element of the manufacturing infrastructure in the context of the fourth industrial revolution. Nevertheless, there is frequently a deficiency of adequate training facilities for future engineering experts in the academic environment. For this reason, this paper describes the development and implementation of two different layer architectures for the metal processing environment. The first architecture is based on low-cost but resilient devices, allowing interested parties to work with mostly open-source interfaces and standard back-end programming environments. Additionally, one proprietary and two open-source graphical user interfaces (GUIs) were developed. Those interfaces can be adapted front-end as well as back-end, ensuring a holistic comprehension of their capabilities and limits. As a result, a six-layer architecture, from digitization to an interactive project management tool, was designed and implemented in the practical workflow at the academic institution. To take the complexity of thermo-mechanical processing in the metal processing field into account, an alternative layer, connected with the thermo-mechanical treatment simulator Gleeble 3800, was designed. This framework is capable of transferring sensor data with high frequency, enabling data collection for the numerical simulation of complex material behavior under high temperature processing. Finally, the possibility of connecting both systems by using open-source software packages is demonstrated.

## 1. Introduction

Since the beginning of the fourth industrial revolution, a paradigm change within the manufacturing environment can be observed [1,2,3,4,5,6]. As an integral part of this revolution, the Reference Architecture Model Industry 4.0 (RAMI 4.0) was introduced [7]. RAMI 4.0 is an extension of the Smart Grid Architecture Model (SGAM) to meet the initial requirements of Industry 4.0 [8,9]. Within this model, information type, system hierarchy as well as asset lifecycle is considered within an administration shell, responsible for the communication between these sections [10]. The inclusion of these key factors is especially important for the development of a smart factory [8,11]. This kind of abstract reference model for layer architectures is not a new concept [12,13,14], but it has a superior advantage due to international standardization. The high amount of the current literature regarding layer architectures demonstrates the importance of this topic among different disciplines in the manufacturing environment, e.g., in [15], Zyrianoff et al. focused the implementation of layered internet of things (IoT) solutions for the development and further enhancement of smart agriculture and smart cities; in [16], Ungurean and Gaitan describe a further concretization of the reference model with a special focus on industrial internet of things (IIoT) solutions; in [17], Gonzalez et al. present the utilization of Modbus TCP to overcome proprietary automation solutions for smart microgrids in the photovoltaic sector. Despite the high academic as well as industrial research activities within the last years [1,2,18], numerous new concepts and developments are not suitable for small and medium sized enterprises (SMEs) operating in the manufacturing environment [19,20]. High investment costs, a high level of standardization in conducted processes (e.g., by lean management approaches) as well as advanced internal IT and data management/digitalization know-how is required for a majority of solutions recommended in the literature [21,22,23,24,25,26,27]. The vast majority of high specialized SMEs do not fulfill these requirements because they have a huge variety as well as low volumes within the production plans. Another characteristic of these businesses is a lower degree of process automation, combined with generally less standardized process management [28,29]. Nevertheless, the economic contribution of SMEs in this sector is not negligible and provides employment opportunities for many current and future graduates of academic institutions [30,31,32]. To ensure sustainable economic development in these companies, variable low-cost digitalization solutions can add major advantages [33,34]. Therefore, interdisciplinary expertise from current and future employees is required in order to achieve this objective [35,36,37]. For this reason, an academic smart factory environment [38] was developed, which serves students and therefore future experts as a practical learning environment to deepen their knowledge in digitalization technologies. In comparison to similar learning factories [39,40,41,42,43,44], the framework discussed in this paper has the advantage of consideration of real physical processes and material parameters (e.g., the possibility of integrating numerical simulation, prediction of microstructure of examined specimens). Furthermore, it supports SMEs by demonstrating low-cost possibilities of digitization and digitalization approaches within the metal processing industry. Despite the hardware solutions, the usage of open source and, more importantly, highly integrative software solutions is of crucial significance. Furthermore, the effort of learning, implementing and updating of such a programming environment must be reasonable. For this reason, Python (Version 3.8) was chosen for the majority of data processing operations described in this case study, using the open source PyCharm Integrated Development Environment (IDE). Python’s increasing popularity in the manufacturing as well as academic world was an additional driver for this decision [45,46]. In addition to the free availability as an open-source product, the increasing popularity is due to the multitude and diversity of the frameworks and their continuous improvement and expansion. Popular frameworks such as pandas enable the preprocessing and manipulation of data [47], Matplotlib visualizes the data [48], and Numpy as well as Scipy allow the elaboration of mathematical operations and machine learning algorithms [49]. Additional frameworks permit the fast assembly of versatile GUIs, e.g., PyQt [50].

For the development of a smart factory layer architecture, efficient and effective data management is key. Digital data storage allows a more efficient, secure and accessible data administration and preservation. Databases are practical for storing and managing data and facilitating the retrieval of specific information. In addition, many databases determine which people or programs can access data depending on the respective permissions. In order to facilitate such a permission system, a database management system (DBMS) is used. For this case, the Structured Query Language (SQL)-based relational DBMS MySQL (Version 8.0.23) was chosen because it is an open source product exhibiting a high compatibility with Python and is simple to learn for engineering students [51]. Furthermore, it provides a straightforward connection to Hypertext Preprocessor (PHP), another widely used open source language for the development of advanced Web applications [52]. Additionally, the hosting can be outsourced to an external server provider or done on in-house servers.

Because there is no all-encompassing solution available for the implementation, suitable for the majority of entrepreneurs, two different layer architectures were developed, depending on the existing IT-infrastructure as well as degree of automation within the respective machine systems. Despite retrofitting approaches, which involve a major proportion of old machine systems with a poor degree of automation, the integration of state of the art machines that already possess a specific digital interface into a not-proprietary IT-framework is of utmost importance [34]. A lot of these systems do not exhibit a standardized open source interface, leading to highly functional, but in most cases isolated, applications [53]. Because the full potential of digitalization and digital transformation lies in the integration of these stand-alone solutions, machine manufacturers commonly offer high cost solutions for the coupling of their individual data acquisition (DAQ) system with other foreign applications [54]. Especially for small and medium-sized enterprises, it is common to avoid these cost-intensive solutions by independently developing efficient solutions.

The following work shows two possibilities of methods capable of gathering and processing necessary data for condition monitoring, maintenance interval optimization and machine learning approaches for engineering education purposes. A special focus lies on the integration of different heterogeneous interfaces as well as easy-to-use human machine interfaces (HMIs) [55,56,57]. Another important attribute of the presented layer architectures is the resilience regarding a harsh manufacturing environment, achieved with the inclusion of data mirroring and strict access right policy [58]. The possibility of adding new layers, e.g., real time numerical simulation as well as a possible interface to an enterprise resource planning (ERP) system was additionally considered.

## 2. Transdisciplinary Engineering Education 4.0: Target Groups and Learning Outcomes

As a result of the fourth industrial revolution and corresponding digitalization and digital transformation in the metal processing environment, required competencies and skills for engineers in this field have changed significantly [59,60,61]. The increase in inter- and transdisciplinary skills necessary to work within this digitalized manufacturing environment must substantially affect the curricula of traditional secondary and tertiary engineering education in order to ensure long-term employability [62,63,64]. For this reason, a new transdisciplinary lecture at the Montanuniversität Leoben was designed. This lecture aims to introduce engineering students of different disciplines into the fundamentals of digitalization and digital transformation in the metal processing environment. Table 1 gives a general overview about affected disciplines at the academic institution.

Students of industrial energy technology, mechanical engineering, industrial logistics, recycling and process technology are heavily affected by the changes in the process and production environment. Therefore, fundamentals of smart-factory-related layer architectures are mandatory for their future careers. Material scientists additionally need to be aware of digitalization in the research and development field. This especially includes know-how about technology-enabled advances in material testing and how this discipline can profit from recent Industry 4.0 related technologies and corresponding advances in sensor technologies. Metallurgists and materials-science-interested mechanical engineers should be aware of developments in both sectors mentioned.

As an integral part to fulfill these requirements, two different layer architectures were developed. The first development focuses on the fundamentals of digitization and digitalization and is based on a low-cost layer architecture, often used in an SME environment (Section 3). Additionally, to point out the importance of such a framework for material scientists, mechanical engineers and metallurgists, the possibilities of including complex FEA into this architecture is elaborated in Section 5. To also demonstrate the potentials and advantages of higher frequency measurement methods for material testing and characterization, a second layer system including fiber optic measurement technologies is implemented on a state-of-the-art thermomechanical treatment simulator. Both architectures transmit data by the Modbus TCP/IP protocol widely used in industrial practice to the internal server system.

## 3. Digitalization and Low-Cost Layer Architecture: Structure and CNC-Lathe Integration

The DAQ is performed by a Wago PFC200 G2 2ETH RS controller, which executes PLC control tasks and internally processes analog and digital signals supplied by input/output (I/O) modules. The I/O modules used are analog input modules that receive analog signals from the CNC-lathe and forward them to the controller in order to convert these analog signals into digital ones that are required for further computer-aided processing (Figure 1).

By connecting three-phase currents measured by a current transformer as well as voltages, the Wago 750-494 analog input module, a three-phase power measurement module, enables real-time measurement of reactive power, apparent power, active power, energy consumption, power factor, phase angle and frequency. The corresponding circuit diagram from the power module point of view is visualized in Figure 2.

Figure 3 shows the implemented measurement and DAQ module. The selected controller is further capable of storing data directly on a SDHC device, serving as an additional security layer. If network transfer would fail, e.g., due to a server maintenance or other, nonplanned downtimes, the processing data is still automatically stored within the memory device.

While the analog module automatically stores the measurement data, additional measurements can be manually added for the purpose of calibration or further specific analysis of defined indicators (e.g., with higher frequency). These measurements can be started and stopped with a graphical user interface (GUI), (Figure 4, dash button ‘Electrical Measurement’), created with the Wago e!Cockpit software suite, which, moreover, allows real-time monitoring of the system parameters.

To apply various data processing programs to acquired data and minimize storage space to a reasonable size, all signals are converted and saved as pre-sorted text-files by an automatically working data transfer protocol, running simultaneously on two local computers. The SD-memory is checked for differences between its storage and the server storage every 24 h. If a deviation is detected (more/different data on the SDHC in comparison to the local raw data file storage), the raw data will be overwritten. In order to avoid a malfunction in the SDHC device, the stored raw data on the server is automatically mirrored, enabling the administrator to investigate potential errors after their occurrence. Because space on the memory card is limited to 32 GB, the card is automatically cleared after exceeding of 80% internal memory space. To guarantee no loss of data, the server storage is mirrored within each 24 h and stored to a SQL database, which operates on a different server partition.

The recorded data set contains the timestamp, active, reactive and apparent powers, currents, voltages, power factors and the quadrants of the three phases (Figure 4, yellow frame). The automatic measurement data recording is realized with a sampling frequency of 2 Hz, which was found to be sufficient from previous evaluations. A preprocessing algorithm also calculates the resulting machine costs according to the consumed apparent power (Figure 4, red frame), serving as a basis for the project management tool.

Figure 5 shows the main GUI for the Python processing layer, which visualizes the non-idle machine hours from the recorded data, analyzed by the embedded programming algorithm [65]. In order to minimize data access time, previously refined data is stored for accounting and general project management purposes in the network within a second mySQL database, and it is made available to technicians and students.

Figure 6 summarizes the first four layers of the low-cost layer architecture for the CNC-lathe, from implemented sensors to the main processing layer.

Table 2 shows the implemented roles and corresponding rights regarding viewing and changing settings within the PHP GUI for an exemplary project. The second SQL database, including the refined data as a result of the main processing layer, serves as an underlying fundamental for this GUI. Within the Python programming environment, input data from the PHP GUI (e.g., new projects or involved coworkers within a specific project) is stored automatically within the refined SQL-database. For the education of engineering students, the developed PHP GUI was duplicated and set up with realistic values to enable a comprehensive experimental setup without disturbing the workflow of respective employees. For this replica, students receive logins for every role, thus enabling them to work with existing roles and corresponding rights. This approach also enables the possibility to change the underlying logic in the back-end of the project management tool, giving deeper insights into PHP-based programming.

The visualization and publishing of refined data for the interactive project management tool is done within the internal network using PHP programming, with a special focus on IT security due to the implementation of different roles with different rights within the PHP GUI (Figure 7).

Figure 8 illustrates the resulting six-layer architecture. To sustain a resilient, adaptive and smooth working system, the machine park and corresponding machine sensors are divided into different nodes. The number of machines coupled to one node is depending on the number of sensors and therefore data transferred, as well as the frequency required. For node 1, two heterogeneous aggregates are coupled to controller 1, whereas the CNC-lathe transfers 25 different indicators with a frequency of 2 Hz, running continuously. This results in a low and steady CPU usage on the respective controller. The second aggregate submitting data through node 1 is a retrofitted cold rolling mill, which transfers data from four different sensors with a frequency of over 500 Hz when operating. This frequency is only achievable through writing data directly on the RAM of the controlling device, resulting in a temporary additional CPU load of more than 80% on the controlling unit. This load peak must be considered when planning digitalization solutions because an overload cannot be avoided persistently in most of low cost controllers. In this case, the necessary algorithm, programmed in structured text format, must also be implemented separately as the used controller initially merely provides up to 1 Hz of acquisition frequency.

## 4. Data Gathering for Initial Condition Monitoring and Further Analysis: A Case Study

As data science fundamentals become more important for future manufacturing experts, a simple case for the reproducible DAQ was defined and carried out. The objective of this approach was to collect data sets which can be easily edited by students on a basic level. Additionally, a simple state-of-the-art logic was implemented, serving as a basis for more sophisticated programming efforts within a supportive learning environment. The respective logic is initially able to distinguish between three states of the lathe system:Off;On but not working (idle time);Working (real machining time).

To be able to differentiate between real machining time (the CNC-lathe operates on a workpiece) and idle machine time (e.g., calibration, adjustment between two machining steps, set-up times), a pretest was carried out. In this pretest, idle mode, tool changer movements and main spindle rotations with different rpm without actually operating on a work piece were performed and analyzed to gain knowledge about the behavior of all recorded electrical parameters. This pretest exhibits the advantage of a low time consumption, allowing lecturers to demonstrate the data collection quickly and therefore enhance awareness of the comprehensive matter. Table 3 shows the 15 different settings investigated.

To analyze real machining time, a cylindrical workpiece (alloy steel, type 708M40) with a length of 250 mm and an initial diameter of 68 mm was axially turned at constant speed using a new cutting blade XMGC30 with an infeed of 0.5 mm per process step. An additional testing plan was created, consisting of constant machining parameters and using axial machining operations to reduce the base material in diameter (Table 4). In order to evaluate the influence of cooling on the power consumption of the machine, tests number 20 and 21 were carried out without the usage of the internal cooling system.

During the entire machining process, the measured sensor data was likewise recorded to identify the corresponding test data faster and figure out correlations between the machining time and the measured values. Another advantage of the control scheme is the opportunity to increase the frequency (simultaneously) while processing without interrupting the continuous DAQ process.

The analysis of all different types of electrical indicators shows that the total active power is the best suited indicator to distinguish between idle time and actual working time. Figure 9 shows the performed testing program according to Table 4, visualized using the Python matplotlib.py extension package.

Figure 10 illustrates the total active power of test 13 and 14, a rotation of the main spindle with 4200 rpm. The power consumption does not significantly deviate for clockwise and counterclockwise rotation. For these tests, the average active power consumptions sum up to about 5100 W. This trend is also consistent with the results of paired tests at other speeds, i.e., test 7 and 8, 9 and 10, 11 and 12.

Figure 11 shows the results of the tests with counterclockwise rotation at different speeds. As already shown in Figure 10, the power consumption remains constant after an initial peak. These plateaus increase in magnitude with speed. By analyzing the measurement data with Python, no trivial correlations or patterns were determined by reactive power, apparent power, phase current, phase voltage, power factor or phase angle. Through the visualization of the total active power, a comprehensible relationship can be established between the machining operation and the evaluated parameters.

As Figure 12 illustrates, a trend displayed by the dashed green line can be observed, which is a representative of the diameter and machined length. The negative measurement peaks, ranging from 50 to 550 W apart from the green trend line in terms of magnitude, represent the tool being set down from the workpiece and returned to the starting position to perform the next programmed process step.

To demonstrate the influence of adequate cooling, test numbers 20 and 21 were performed without cooling (Figure 12, red dashed line), resulting in a lower total active power in comparison to other test samples. The constant deviation from the green dashed trend line by an offset in magnitude can be explained as a result of decreasing power consumption due to unused aggregates for coolant supply. The number of negative peaks within Figure 12 is equivalent to the number of process steps for each test.

Figure 13 shows six of these smaller negative peaks that are equivalent to the number of processing steps of test number 16. If negative peaks fall below a total active power of 1500 W, the machinery is not operating—the time during which the total active power falls below this value does not contribute to machining and can be excluded from the machine-hours-counting algorithm.

Table 5 demonstrates the accuracy of the investigated behavior. A wrong classification of machining parameter data points is below 0.03%, which is not significant in terms of maintenance or machine hour calculations, and therefore, it is an activity-cost-based project management approach. If these tests were conducted more frequently, a higher number of heterogeneous datasets would be generated and simple machine learning algorithms for the classification of the respective data (e.g., Support Vector Machines, Decision Tree Analysis) can be instructed [66,67].

Due to the relatively short recording time for test numbers 7 to 14 (20–30 s), higher mean values and standard deviations arise compared to machining tests 16 to 18. The shorter the recording time, the higher the influence of peaks at the beginning and the drop at the end of the data set (Figure 9 and Figure 10), leading to the resulting deviation. This divergence also demonstrates the significance of encompassing statistics behind data-driven technology and the relationship between the amount of data and prediction accuracy.

Table 6 shows the calculated peak values, the mean values as well as the standard deviations of all test numbers listed in Table 3 and Table 4. The precise identification of real machining and therefore actual wearing of the analyzed aggregate has several advantages. Before the development of the discussed architecture, ordinary maintenance was executed after specific time intervals, instead of considering the effective wear of the machine system. The implementation of this framework enables maintenance intervals to be determined on the basis of actual machine hours. This approach leads to lower maintenance costs because unnecessary servicing is minimized and additional necessary maintenance is recommended. As a result, periods with higher machine utilization are identified automatically and quantitatively. For a more efficient scheduling, the residual time until the next external service is calculated as a moving average. The exact predictability increases with the duration of the system’s utilization. For a further cost reduction, a standardized internal calibration test was developed. After exceeding 25% of calculated machine hours until next external service, a standardized test, serving as an indicator for possible malfunctions within the aggregate will be executed. The machining time left until the next internal service is implemented within the project management GUI (PHP/Python) as well as programmed Wago GUI (structured text). After internal or external service, the calculation can be reset within the corresponding GUI.

A substantially more precise calculation can be achieved by the developed project management GUI. As the system provides the real power consumption of the aggregate, internal as well as external projects can be calculated on a more reality-based manner. While the PHP interface authorizes respective project leaders to set up new projects and enter personnel costs with or without the usage of machines, the system also substitutes different manual working hour recordings, which were carried out for internal projects individually and more qualitatively until the implementation.

To ensure a learning experience that is as close to reality as possible by a reasonable data set from machine systems as well as the developed project management tool, the initial framework is also used on a daily basis by the personnel at the institution. When implementing new IT infrastructures with a higher level of automation, it is essential to involve the staff and identify their preferences at an early stage of the introduction. Therefore, all co-workers were briefed and asked about their opinion towards a project management system and what it is supposed to contain to facilitate the daily workflow. As a result, the PHP GUI was adapted several times, considering the preferences of respective employees. Moreover, the Wago as well as PHP GUI is available on every computer device within the local network of the academic institution, which allows all involved personnel to start measurements, overview specific machines and create or update projects independently from their specific location (depending on individual rights). Through secured VPN access, a completely remote condition monitoring is possible. This degree of freedom also offers students the possibility to engage with and refine the system remotely if access is given by respective lecturers.

## 5. Integration of Numerical Simulation and Implementation of a High Frequency DAQ Architecture

Due to the rise in computational capacity and speed within the last decades, the possibility of integrating real-time numerical simulation within the actual production process becomes more and more suitable among the manufacturing environment [68]. Therefore, the presented framework can be extended to include numerical simulation (near) real time in a variety of production processes.

Figure 14 visualizes the additional integration of a finite element analysis (FEA) program within the developed framework. In this example the Python GUI adapts different rolling steps within one rolling operation based on the results of a FEA, calculated during the time required for the previous process within the production operation and under consideration of processing and material properties.

Based on the knowledge gained from the case study in Section 4, the integration of sophisticated numerical simulations into the framework derives in a broader understanding of the possible advantages of these technologies. Nevertheless, most material processing operations, especially high temperature forming processes, require constant surveillance of the material behavior under enhanced temperature and forming conditions. In order to be able to handle a forming process of a particular material, it is necessary to have a certain comprehension of microstructural changes. In general, extensive material parameter studies are indispensable for predicting the final microstructure resulting from the forming process, such as anisotropy and the resulting grain size or grain size distributions, as well as possible material damage influencing variables [69].

In an increasing number of cases, integrated microstructure models are used in the numerical simulation of a forming process as accompaniment, relating the occurring forming parameters (e.g., temperature gradient, strain rate) to the resulting microstructure changes such as static or (meta-) dynamic recrystallization as well as grain growth [70]. The required material parameters are commonly obtained in suitable thermomechanical simulators, that operate on a laboratory scale [71,72,73,74,75]. Since the processes proceed expeditiously, especially in the case of simultaneous forming at high strain rates, it is essential for material data acquisition to ensure a significantly higher sampling frequency of the system [76,77]. For this reason, an additional DAQ system provided by iba was implemented at the institution’s thermo-mechanical treatment simulator (Type Gleeble 3800). This DAQ is widely used in industrial practice, offering different software packages and the possibility of significantly higher sensor sampling rates for further processing [78,79]. The connection between the sensors and the system is realized with a proprietary A/D converter, transferring digitized data by a fiber optic line with up to 100 kHz on four channels:Temperature;Dilation of the respective specimen;Resulting Force;Displacement.

The gathered data is preprocessed directly within the ibaAnalyzer software package and automatically submitted to a file system hosted by the internal server architecture of the institution.

The high sampling rate offers the possibility of investigating the influence of time-dependent changes in material behavior by measured values. The resulting data sets can be further used to develop and adapt numerical models to digital shadows and, in long instances, digital twins [80,81].

The Gleeble system, like a majority of highly specialized material testing aggregates, offers a proprietary software solution for resulting data analysis. By recording a hot tensile test of bainitic steel and comparing the results of both data sets, previous work of the authors revealed a significant difference in the gathered temperature data, which indicates an internal data preprocessing and correction of the proprietary software unit [38]. Due to the low output voltage signal of thermocouples used, a voltage fluctuation within 10^−3^ V results in a temperature deviation of 250 K. Figure 15 illustrates this deviation. These examples can be used to raise awareness about these kinds of potential inaccuracies.

Figure 16 visualizes the resulting architecture, from the applied sensors to the (refined) data storage at the internal server.

## 6. Results and Discussion

As a result of this work, a six-layer architecture was designed, concentrating deliberately on the use of a few selected products (open source if possible) in order to make application and modification by interested parties as simple as possible. Additionally, a second DAQ system was developed to give this group the opportunity to gain data for real physics-based numerical simulations. Besides, the most important objective was to create a smart factory layout that enables students and practitioners from the metal processing field to engage with different levels of digitization and digitalization, reaching from analog signal to numerical simulation integration. The resulting layer architecture is highly adaptable in terms of the used programming languages (e.g., Python can be substituted with C++ or Java if preferred; mySQL can be substituted with flux). This architecture fulfills three purposes. First, a technical fundament for teaching students in manufacturing related disciplines was created, which allows the following:To gain an overview about the most important fundamentals of networking technologies and corresponding protocols in the manufacturing environment;To deepen knowledge on manufacturing related data science by working with different amounts and homogeneous as well as heterogeneous data sets;To be able to work with different types of DAQ systems used in industrial practice;To optimize interfaces and investigate interface-related efficiency and effectivity concerns in-person or remotely;To enhance knowledge about common programming languages and machine learning technologies in manufacturing by working with real data from machining processes;To obtain an overview of interactive project management and how (near) real-time adaption of required parameters (e.g., cost changes) can affect project outcomes;To raise awareness about the importance of transdisciplinary communication and education in the manufacturing field.

The second operational area of the implemented framework is the research and development of state-of-the-art digitalization technologies, based on this initial work by the following:Extending the framework with other, more complex machine systems (e.g., hydraulic presses, ovens);Extending the framework with more complex machine systems by developing predicting algorithms including thermo-mechanical properties of materials;Using this algorithms for the transformation of existing machine systems to Cyber Physical Production Systems (CPPS) based on the brownfield approach [82,83];Integrating further open-source-based logic between these CPPS, resulting in a superordinate Cyber Physical Logistic System [84,85,86].

The third purpose is the collaboration with interested parties from the industry, especially SMEs, who can use this framework within interdisciplinary projects. This approach has the main advantage of giving industrial experts the opportunity to deepen their knowledge or perform highly experimental tests. Additionally, engineering students are given the possibility to collaborate with these companies from an early stage, gaining additional practice and establishing networks already during their studies.

The presented architecture in Section 3 is an efficient and effective way of taking advantage of current information and communication technologies within a small volume and high-variety production environment. The tools and programs used are either low-cost or even completely free-of-charge, therefore providing an ideal basis for digitalization of small production facilities from scratch. To build up such a low-cost, resilient system, the following points must be considered:How many different channels (different values from sensors, e.g., pressure, force, dilation, temperature) are needed for each respective machine system? (specification of needed input modules);Which frequency is needed for each channel? (avoidance of aliasing, dependent on the process and respective material characteristics);What kind of database is applicable within the respective company? (considering internal know-how and experience);How resilient does the physical hardware and software have to be? (dirt, dust, temperature, accessibility, space);What IT-infrastructure serves as a basis for the framework? (Windows, Linux, other server—OS);What kind of GUI/HMI do respective employees favor?

The individual answer to (1) implies knowledge about all respective machine systems. In general, one can recommend starting with one system where all (from a present point of view) required sensors are already applied and the resulting data is understood.

Answering (2) seems more complex because the required frequency depends on the purpose. In the case of the rolling mill at the academic institution, a medium frequency is needed. In case of the discussed CNC lathe, a much lower frequency is applicable because the process itself is highly standardized trough the internal machine control unit. For high temperature or high-speed forming processes, a significantly higher sampling rate has to be ensured. In general, if the material behavior itself should be analyzed, higher frequencies are mandatory (e.g., considering microstructural changes due to applied or internal forces or as a function of the temperature gradient in case of an involved heat treatment).

Question (3) is dependent on the internal knowledge. If no specific database system is used, open-source programs can be recommended.

Question (4) is heavily dependent on the specific environment. If existing sensors are working within the environment, the sole important point to consider in this case is the resilience of the respective controller. Most Supervisory Control and Data Acquisition (SCADA) suppliers offer specific, more robust solutions (e.g., Wago XTR series).

Regarding (5), an efficient and stable interface between the resulting storage solution (server or PC) must be programmed. In this study, a regular windows system was used. One of the advantages that Python and its various extension packages offer is the very broad possibility of interface programming. There are different types of extensions for the coupling of different IT-systems to the controller system available. The controller system itself in this case produces txt-files, which then were automatically implemented in the SQL based database system as well as stored parallel on the used windows server system.

The answer to (6) is crucial for a successful implementation. Without considering the experience and preferences of involved employees on the shop floor, a well-planned digitalization solution is likely to fail. Including respective workers in the development of user interfaces at the earliest possible stage helps to successfully implement and sustain the change in working environment.

The second architecture should serve as an additional expansion to higher frequency DAQ technologies with a special focus on data gathering for numerical simulations. From a network technology and data science point of view, the most essential questions to answer, additionally, are as follows:7.What sampling rate is sufficient to obtain enough data for an accurate material behavior prediction? (e.g., recrystallization behavior of the investigated material under defined process parameters)8.How accurate are implemented DAQ systems? Is it possible to confirm resulting data?

These points should be considered intrinsic by each engineering student who strives for a career in a digitalized metal processing environment. This work should therefore give an experimental basis to concretize the answers given by the author for specific cases.

## 7. Conclusions and Outlook

This paper describes the development of a six-layer smart manufacturing architecture for the transdisciplinary engineering education. For this purpose, two DAQ systems—one to demonstrate fundamentals and possibilities of open-source low-cost digitalization solutions and a second for high frequency measurement applications—were developed and implemented. For both architectures, case studies were provided to enhance comprehensible teaching in a digitalized manufacturing environment. A major advantage of the proposed structure is the open-source components used wherever possible. The selected technologies are already common in industrial practice, due to the high degree of connectivity, cost efficiency and practicability in the metal processing environment.

As measurement results of the high frequency architecture are stored within the same server architecture as the Wago DAQ system, respective data can be analyzed and further processed in the same Python environment. The Gleeble system coupled in this network is also a widely used simulators in the industrial practice, especially in the research and development field. By including this system into the layer architecture and coupling this architecture with a superordinate MES, the horizontal integration of different departments in the manufacturing environment can be simulated. Because the complexity in the academic institution’s learning factory (14 heterogeneous machine systems with different initial degree of automation) can be defined as similar to those in SMEs, a low-cost open-source solution can be programmed and implemented to serve as MES. By using Python for this purpose, already-existing extensions for the coupling with an ERP program can be realized efficiently, allowing students and future manufacturing experts to use this framework for the simulation of manufacturing processes from initial digitization to the coupling with, e.g., corporate accounting or procurement. The Montanuniversität Leoben additionally launched the new bachelor’s program Industrial Data Science, focusing on the transdisciplinary engineering education with special emphasis on data gathering and processing within the material processing environment. As additional machine systems are integrated within the frameworks, machine learning algorithms can be further implemented and optimized by interested engineers for data monitoring applications. The monitoring and malfunction detection as well as related IT-security issues, highly discussed in the current literature [87,88,89], can further be used for the deeper education of future industrial data scientists.

## Figures and Tables

**Figure 1 sensors-21-02944-f001:**
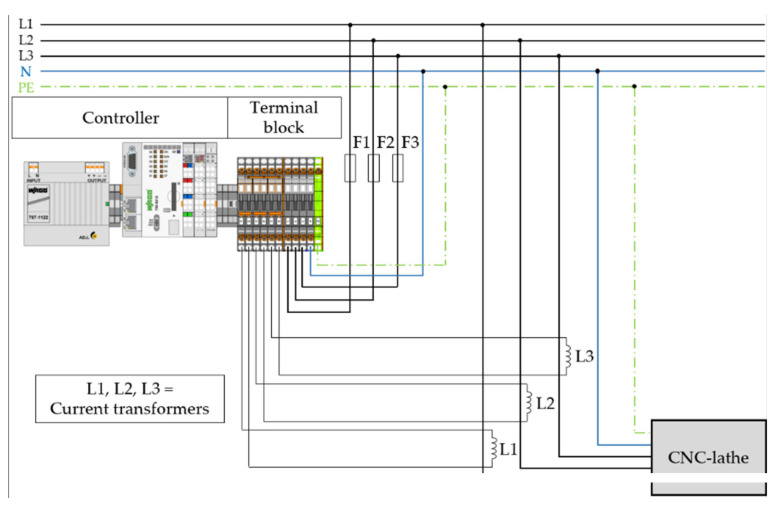
Circuit diagram for the connection of the CNC-lathe to the superordinate system.

**Figure 2 sensors-21-02944-f002:**
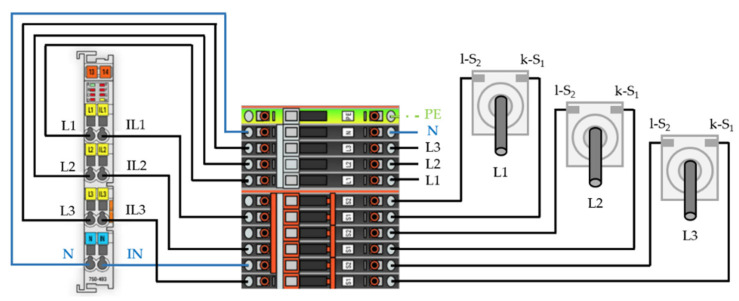
Circuit diagram: power module side.

**Figure 3 sensors-21-02944-f003:**
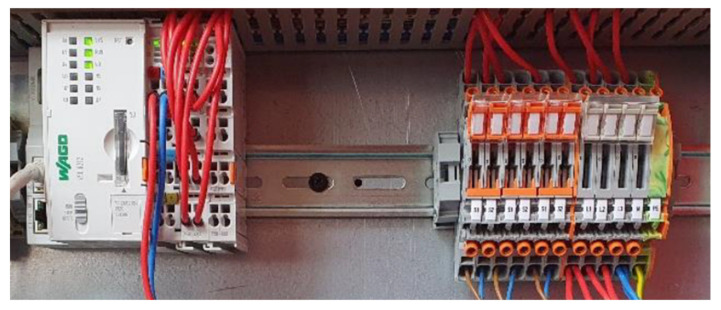
Controller (left) and terminal block (right) with wiring.

**Figure 4 sensors-21-02944-f004:**
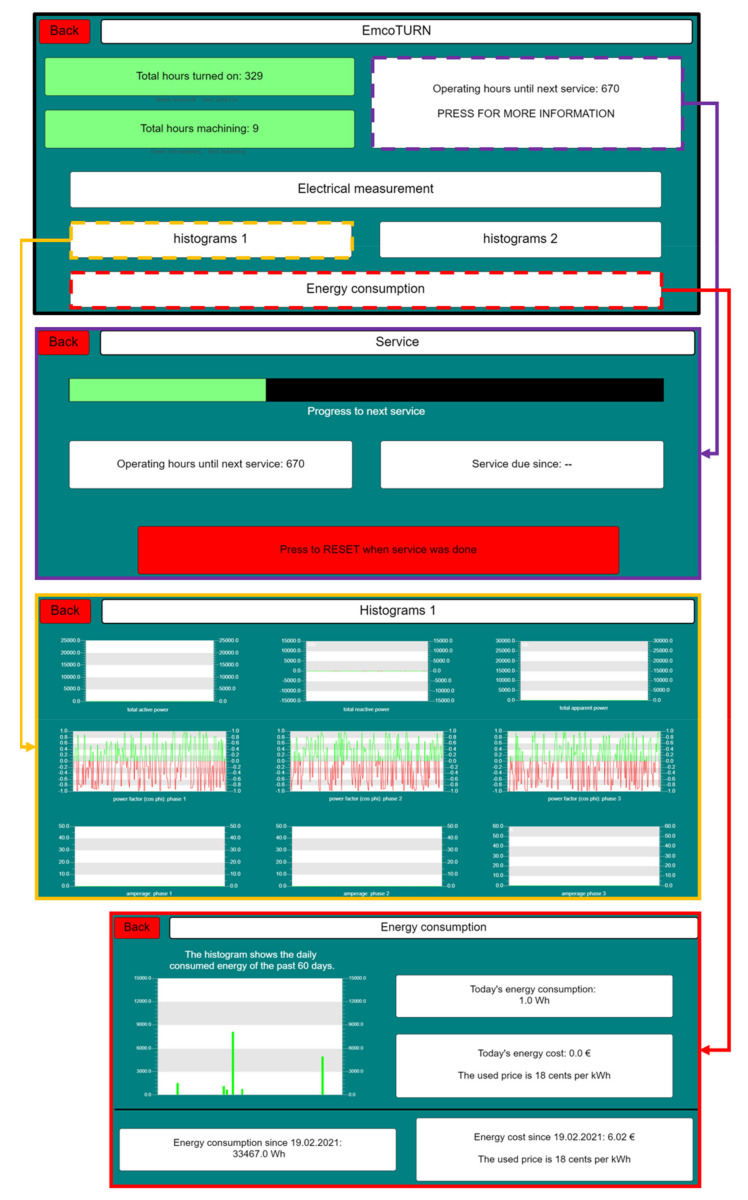
Wago GUI for measurement control (CNC-lathe).

**Figure 5 sensors-21-02944-f005:**
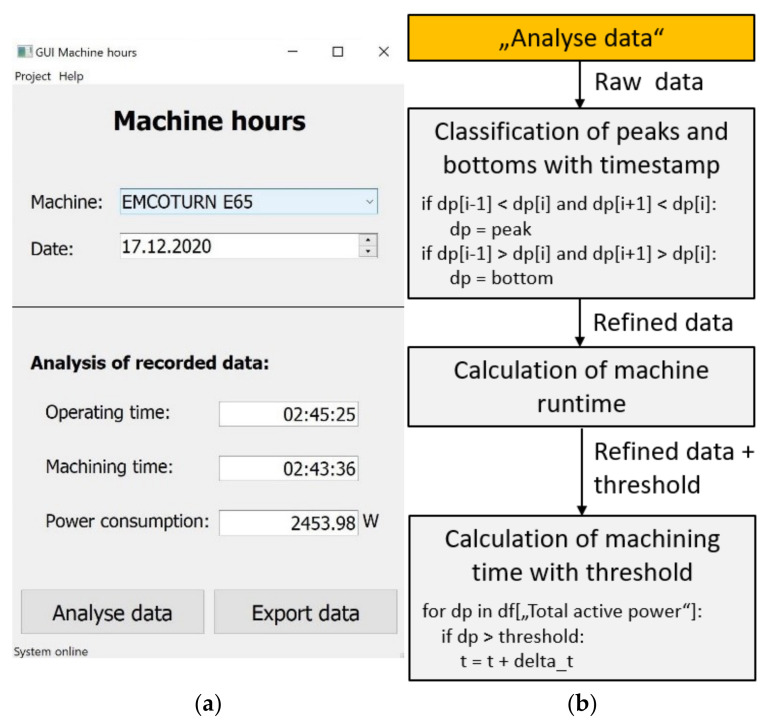
Python logic for machine hour counting: (**a**) visualization, programmed in QtPy; (**b**) back end logic for the GUI.

**Figure 6 sensors-21-02944-f006:**
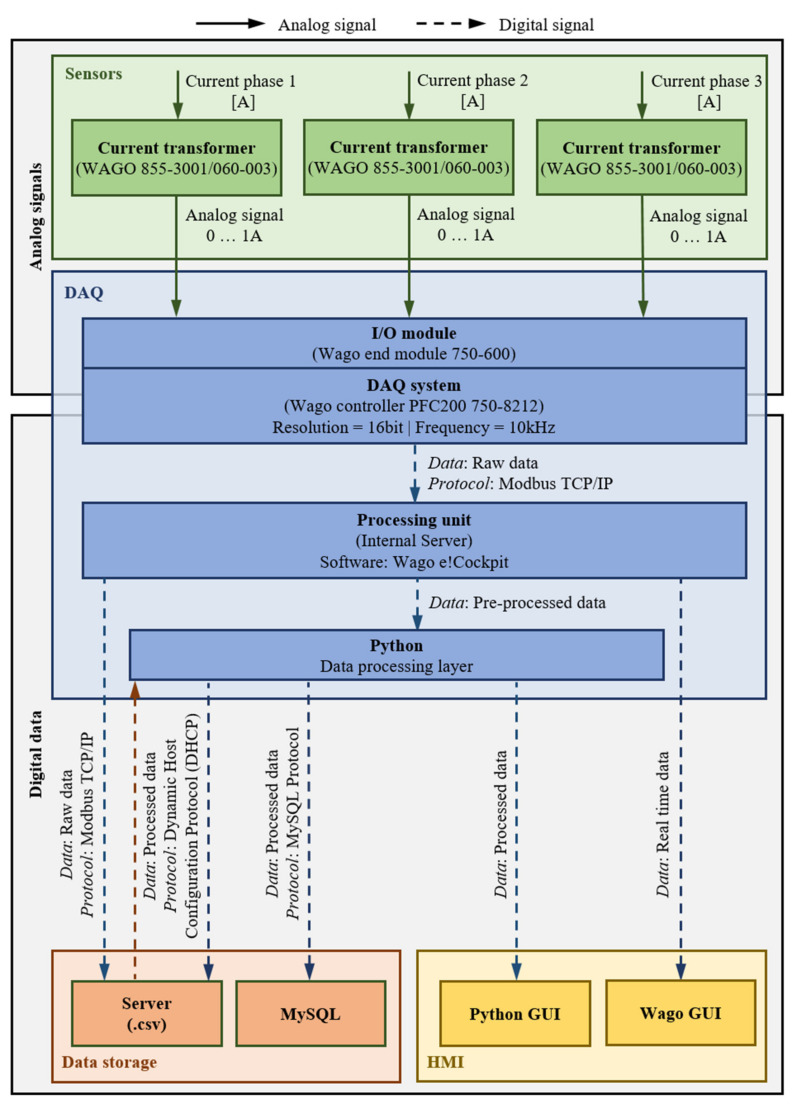
Data flowchart for the integration of the CNC-lathe into the low-cost layer architecture.

**Figure 7 sensors-21-02944-f007:**
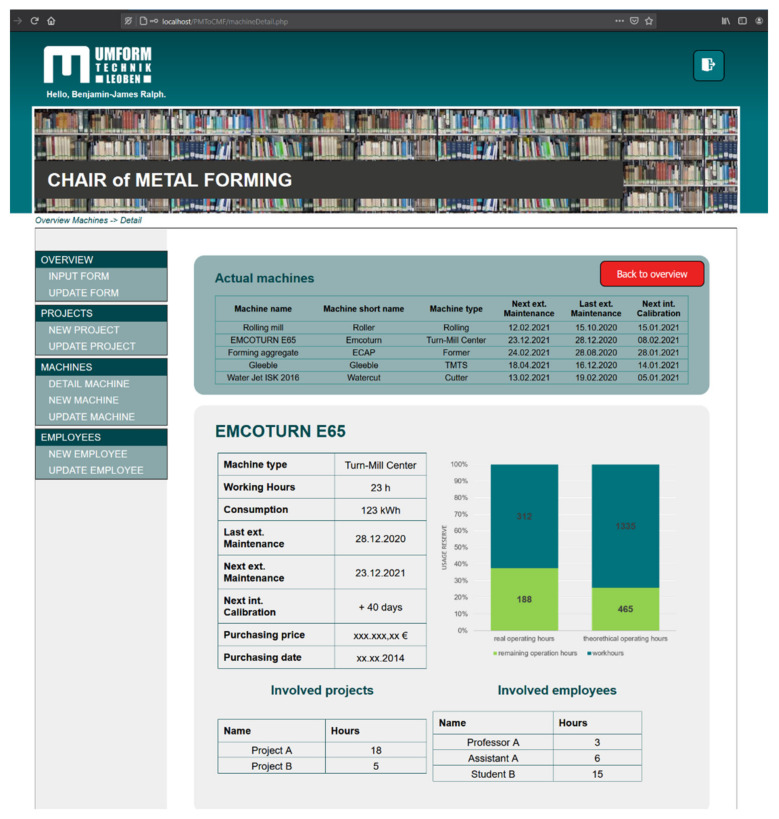
Interactive project management tool, programmed in PHP and directly coupled to an underlying MySQL database. The SQL database is coupled within the Python logic presented in this work.

**Figure 8 sensors-21-02944-f008:**
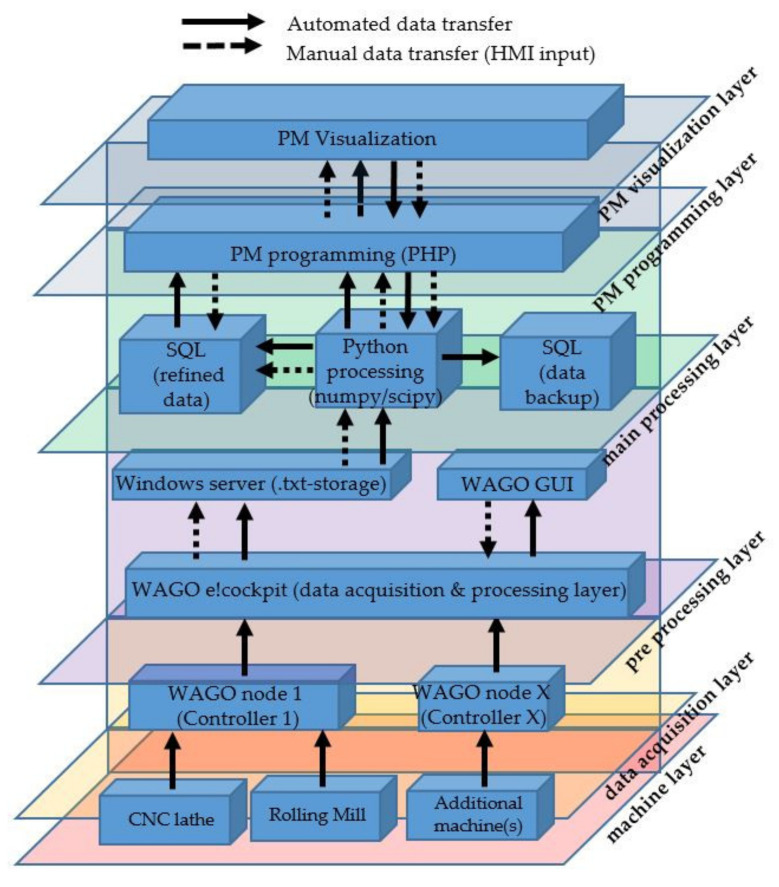
Resulting low-cost six-layer architecture.

**Figure 9 sensors-21-02944-f009:**
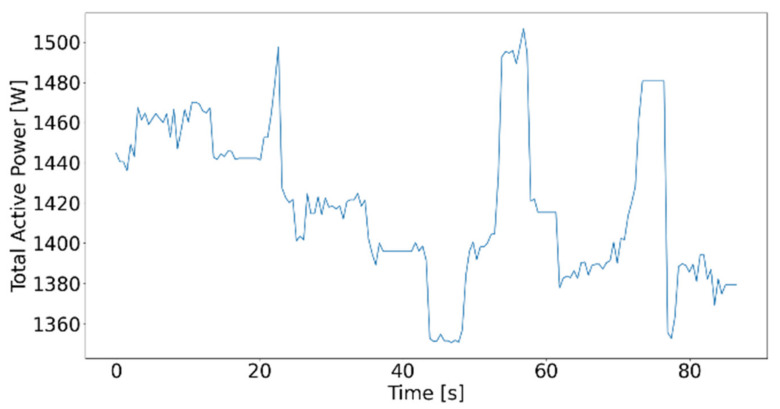
Total active power of idle tests carried out.

**Figure 10 sensors-21-02944-f010:**
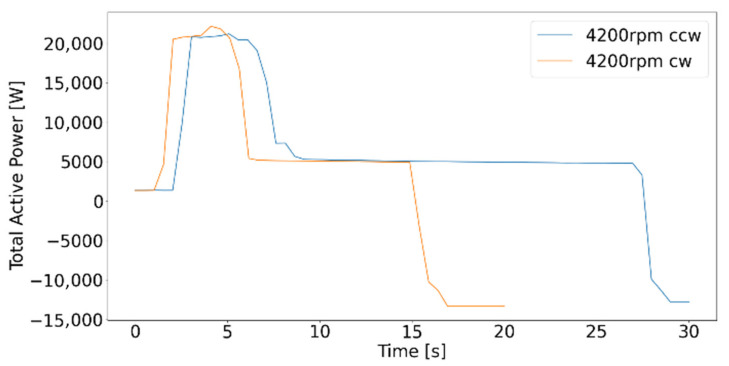
Total active power of tests 13 and 14.

**Figure 11 sensors-21-02944-f011:**
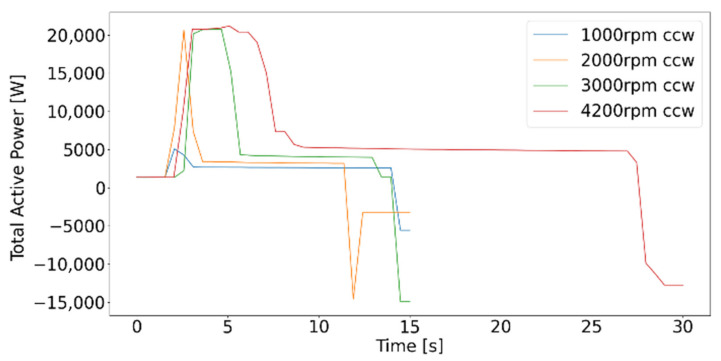
Total active power of tests 7, 9, 11 and 13.

**Figure 12 sensors-21-02944-f012:**
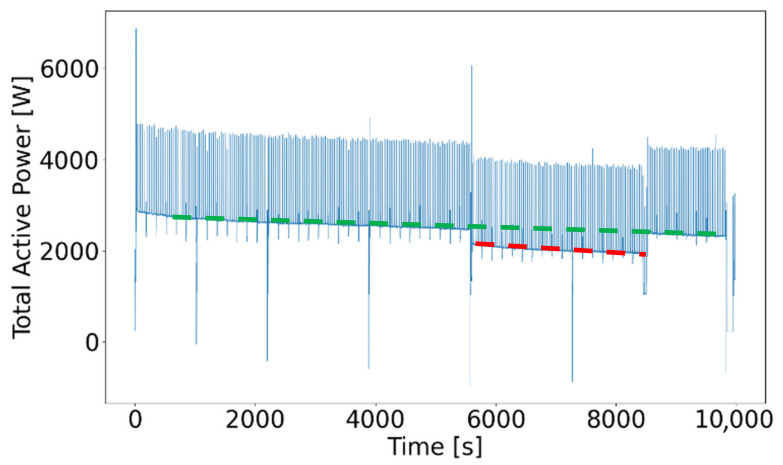
Total active power during axial machining.

**Figure 13 sensors-21-02944-f013:**
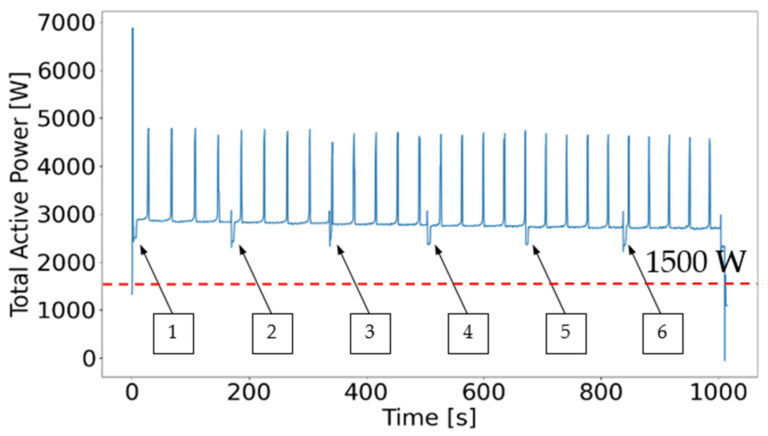
Total active power of test number 9.

**Figure 14 sensors-21-02944-f014:**
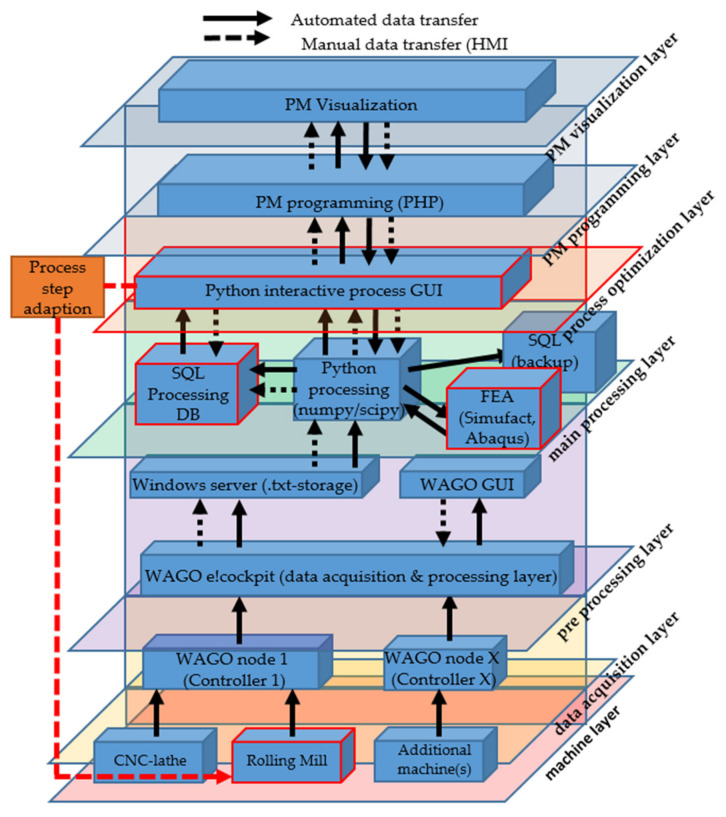
Six-layer architecture with integrated numerical simulation: FEA digital shadow for semi automatized process adaption (example rolling mill).

**Figure 15 sensors-21-02944-f015:**
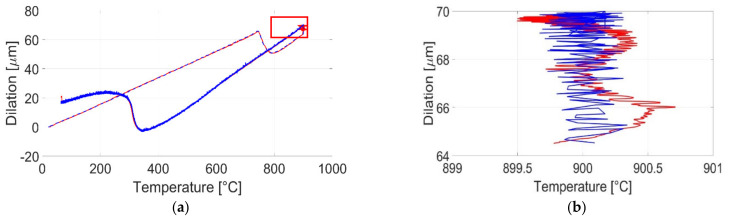
Dilation curve for a tensile test of bainitic steel, carried out with the Gleeble system [38]: (**a**) temperature change with respect to dilation, blue line: Gleeble data set, red line: iba data set; (**b**) cutout area of deviation between both data sets from (**a**).

**Figure 16 sensors-21-02944-f016:**
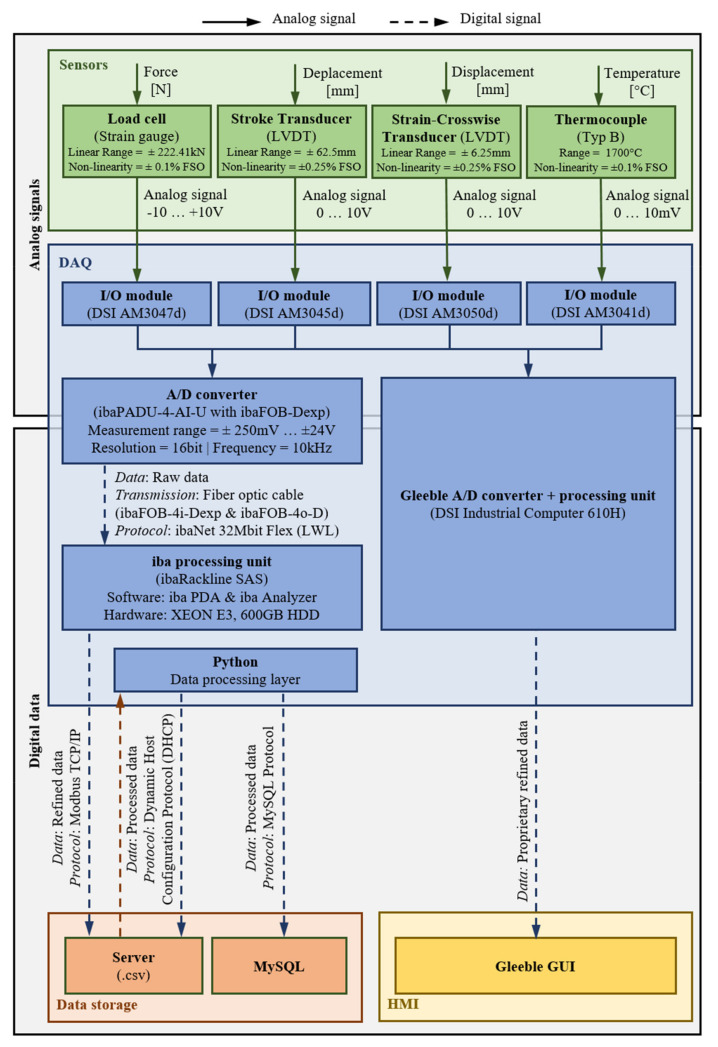
High frequency DAQ and data storage.

**Table 1 sensors-21-02944-t001:** Main target engineering disciplines at the Montanuniversität Leoben.

Engineering Focus	Associated Programs at the Montanuniversität Leoben
Energy	Industrial Energy Technology
Materials	Materials Science
Process and Product	Metallurgy; Mechanical Engineering; Industrial Logistics
Recycling	Industrial Environmental Protection and Process Technology; Recycling

**Table 2 sensors-21-02944-t002:** Project management GUI: implemented roles and corresponding rights (E = employees/M = machines/P = project).

Role	Admin	Project Leader	Project Member	Technician	Other Personnel
Overview	X	X	X	X	X
Detail view	E/M/P	E/M/P	M/P	M	-
Set new project activities	X	X	-	-	-
Budget & cost details	X	X	-	-	-
Employee details	X	X	-	-	-
Change milestones	X	X	-	-	-
Change budget	X	-	-	-	-

**Table 3 sensors-21-02944-t003:** Testing program for the identification of idle related change in electrical indicators.

Test No.	Type of Testing
1	X− transition of tool turret
2	X+ transition of tool turret
3	Z− transition of tool turret
4	Z+ transition of tool turret
5	Z− transition of tailstock
6	Z+ transition of tailstock
7	Counterclockwise rotation with 1000 rpm of main spindle
8	Clockwise rotation with 1000 rpm of main spindle
9	Counterclockwise rotation with 2000 rpm of main spindle
10	Clockwise rotation with 2000 rpm of main spindle
11	Counterclockwise rotation with 3000 rpm of main spindle
12	Clockwise rotation with 3000 rpm of main spindle
13	Counterclockwise rotation with 4200 rpm of main spindle
14	Clockwise rotation with 4200 rpm of main spindle
15	Full 360°rotation of tool turret

**Table 4 sensors-21-02944-t004:** Calibration plan and parameters for machining.

Test No.	Initial Diameter (mm)	End Diameter (mm)	Cooling	Rotational Speed (1/s)	Feed in (mm)	Cutting Speed (mm/s)
16	68.0	62.0	Yes	10	0.5	1.5
17	62.0	55.0	Yes	10	0.5	1.5
18	55.0	45.0	Yes	10	0.5	1.5
19	45.0	35.0	Yes	10	0.5	1.5
20	35.0	25.0	No	10	0.5	1.5
21	25.0	18.0	No	10	0.5	1.5
22	18.0	10.0	Yes	10	0.5	1.5

**Table 5 sensors-21-02944-t005:** Data point classification.

	Data Points Real Machining	Data Points Idle Machining
Sum	19,960	174
Right	19,440	173
Wrong	520	1
% Wrong	0.026	0.0057

**Table 6 sensors-21-02944-t006:** Analysis of peak values, mean values and standard deviation of all tests.

Test No.	Peak (W)	Mean (W)	Standard Deviation (W)
1	1400.52	1373.97	22.45
2	1506.73	14,447.17	41.40
3	1470.10	1453.34	11.17
4	1497.68	1417.77	23.17
5	1480.63	1416.03	39.87
6	1394.20	1379.81	11.69
7	5081.58	2053.05	2198.34
8	10,064.27	2586.07	1864.12
9	20,644.23	1979.91	5508.34
10	20,754.51	2997.70	6094.99
11	20,723.80	4685.71	8261.15
12	20,752.01	607.02	10,746.6
13	21,175.08	5664.18	7751.18
14	22,137.44	3658.69	11,133.24
15	3374.98	1949.28	864.02
16	6879.55	2840.18	403.78
17	4655.05	2731.31	406.98
18	4590.80	2676.75	405.01
19	4929.11	2599.64	423.23
12	6070.17	2140.74	409.52
21	4245.32	2035.68	449.24
22	4556.87	2270.27	717.81

## Data Availability

The data presented in this study are available on request from the corresponding author.

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
