# Peer review of "Implementation of a Six-Layer Smart Factory Architecture with Special Focus on Transdisciplinary Engineering Education"

_sensors, 2021, doi:10.3390/s21092944_

Round 1
Reviewer 1 Report
In the paper, the authors are proposing six-layer smart factory architecture with special focus on transdisciplinary engineering education. Taking into account that smart factories are an integral element of the manufacturing infrastructure in the context of the fourth industrial revolution the theme is very interesting and has potential. The abstract is adequate and precise. The six-layer smart factory architecture is well presented and well-grounded and in my opinion. It will be very useful for the readers as well to better understand the proposed concept. Conclusions are supported by the results. However, in my opinion, there are some issues that should be solved in order to improve the overall quality of the paper.
- The value of the article would be much greater if the authors introduced an extension of discussion elements and comparisons with the results of other studies.
- The main weakness of the paper is the lack of research methodology.
- The paper needs some editorial work e.g. the quality of the figures must be improved. e.g Figure 1, 2, 4.
Reviewer 2 Report
The idea of the paper brings value to smart manufacturing education. However the structure of the paper confuses readers and make the paper difficult to follow. I suggest the authors restructure the paper according to the traditional research paper guideline and reorganize the content so it is clear to readers that why this work is needed, what is the structure of the design, and what are the designs.
Reviewer 3 Report
The topic of the manuscript is interesting given the attention that layered functional architectures are receiving from different scientific fields; in addition, this topic fits the scope of the Journal. After a careful revision, the following comments are provided for the enhancement of the manuscript.
- Regarding the format of the document, some suggestions are as follows.
The acronym GUI should be defined in the Abstract. The same consideration is done for SCADA in line 469.
In the keywords list, the last semicolon is not required.
In line 35, “know how” must be hyphenated.
In line 167, “python” should be capitalized.
In the caption of figure 6, “data base” appears whereas in the rest of the document “database” is used. A unique term should be considered.
References must be slightly revised to fit the format of the template. For instance, abbreviated name of journals must be used.
Sections like Authors contributions and Data availability should be revised since they contain the by default text of the template.
- The paper is well written and organized; about the content of the manuscript, these issues are commented.
“Architecture”, or “Layer architecture”, could be added as keyword, if the authors agree.
The contextualization is well scheduled and presents the topic of the manuscript. However, there is no mention to the necessity of layered architectures as well as to the state of the art in this regards. Hence, this contextualization could be enhanced by including some recent references which deal with layered architectures in order to highlight the relevance of the presented proposal. For example, the following ones can be considered by the authors:
- Ungurean, I.; Gaitan, N.C. A Software Architecture for the Industrial Internet of Things—A Conceptual Model. Sensors 2020, 20, 5603. https://doi.org/10.3390/s20195603
- González, I.; Calderón, A.J.; Portalo, J.M. Innovative Multi-Layered Architecture for Heterogeneous Automation and Monitoring Systems: Application Case of a Photovoltaic Smart Microgrid. Sustainability 2021, 13, 2234. https://doi.org/10.3390/su13042234
- Trunzer, E.; Calà, A.; Leitão, P.; Gepp, M.; Kinghorst, J.; Lüder, A.; Schauerte, H.; Reifferscheid, M.; Vogel-Heuser, B. System architectures for industrie 4.0 applications. Prod. Eng. 2019, 13, 247–257. https://doi.org/10.1007/s11740-019-00902-6
- Zyrianoff, I.; Heideker, A.; Silva, D.; Kleinschmidt, J.; Soininen, J.-P.; Cinotti, T.S.; Kamienski, C. Architecting and deploying IoT smart applications: A performance—Oriented approach. Sensors 2019, 20, 84. https://doi.org/10.3390/s20010084
In addition, the RAMI architecture should also be, at least, briefly commented.
Python is a programming language very powerful and versatile, which is being introduced in research applications during last years. In this sense, given the importance of Python in the proposed architecture, this reviewer recommends including a mention to such features and, even, to the reasons to choose this language.
In a similar sense, MySQL database is mentioned in the caption of figure 6, but the rest of the paper does not comment about such database, only about a SQL database. Once again, given the importance of data storage and, therefore, of such database, it is suggested to briefly comment the selection of such DBMS as well as other design/deployment details (version, hosting, etc.) that the authors consider relevant for the interested reader.
The reference to Figure 4 in line 229 is unclear. Perhaps an additional statement to clarify the content is required.
In line 399, instead of “flux” the authors refer to “InfluxDB”.
How is the DAQ Gleeble connected? Does it use Ethernet? Apart from this particular detail, the authors should mention in a clear manner the communication protocols and networks that are integrated within the proposed architecture (PROFINET, PROFIBUS, MQTT, etc.). This aspect acquires certain importance in real industrial practice where a number of legacy and modern protocols are used.
The reported architecture is envisioned to be used in educational context. Consequently, it would be desirable to find some illustration of the degrees, disciplines and skills where it is focused. For example, a table could be useful to this purpose. It could be placed in section 5.
Regarding such section 5, the title of Results does not seem adequate from this reviewer viewpoint. The architecture in section 4 has been described but not validated. This issue could be partially solved if some data of implementing such architecture were shown, as already done in section 3.
Moreover, the text in the current section 5 is more a discussion that results.
Including a figure within the last section of a paper is not a common practice. Perhaps, it could be placed within the previous section, Discussion, or, even, removed from the paper.
Still concerning the last section, this reviewer suggests highlighting the suitability of the applied open-source tools. This is a positive feature of the reported research and deserves certain emphasis given the increasing utilization of this technology in advanced frameworks.
Round 2
Reviewer 1 Report
Dear Authors, the very much for your responses.
Reviewer 2 Report
It's ok to accept.
Reviewer 3 Report
The new version of the manuscript properly addresses the provided suggestions. Consequently, it has been improved. Congratulations to the authors for their efforts.